# Improving the Stability of Oil Body Emulsions from Diverse Plant Seeds Using Sodium Alginate

**DOI:** 10.3390/molecules24213856

**Published:** 2019-10-25

**Authors:** Yuemei Zhang, Nan Yang, Yao Xu, Qian Wang, Ping Huang, Katsuyoshi Nishinari, Yapeng Fang

**Affiliations:** 1Glyn O. Phillips Hydrocolloid Research Centre, National “111” Center for Cellular Regulation and Molecular Pharmaceutics, Key Laboratory of Fermentation Engineering (Ministry of Education), Department of Bioengineering and Food Science, Hubei University of Technology, Wuhan 430068, China; zym19950721@163.com (Y.Z.); xy15827478521@163.com (Y.X.); yingzi415768596@163.com (Q.W.); huangpipi0601@163.com (P.H.); katsuyoshi.nishinari@gmail.com (K.N.); ypfang@sjtu.edu.cn (Y.F.); 2Food Hydrocolloid International Science and Technology Cooperation Base of Hubei Province, Hubei University of Technology, Wuhan 430068, China

**Keywords:** oil bodies, emulsion stability, sodium alginate, protein-polysaccharide interaction, viscosity

## Abstract

In this study, peanut, sesame, and rapeseed oil bodies (OBs) were extracted by the aqueous medium method. The surface protein composition, microstructure, average particle size $d_{4,\;3}$, *ζ*-potential of the extracted OBs in aqueous emulsion were characterized. The stability of the OB emulsions was investigated. It was found that different OB emulsions contained different types and contents of endogenous and exogenous proteins. Aggregation at low pHs (<6) and creaming at high pHs (7 and 8) both occurred for all of three OB emulsions. Sodium alginate (ALG) was used to solve the instability of OB emulsions under different conditions—low concentration of ALG improved the stability of OB emulsions below and near the isoelectric point of the OBs, through electrostatic interaction. While a high concentration of ALG improved the OB emulsion stability through the viscosity effect at pH 7. The OB emulsions stabilized by ALG were salt-tolerant and freeze–thaw resistant.

## 1. Introduction

Oil bodies (OBs) are natural droplets with a core of liquid triacylglycerols (TAGs) surrounded by a monolayer membrane of phospholipids embedded with OB endogenous proteins [1,2]. These surface proteins include mainly oleosin (15~25 kDa), caleosin (27~30 kDa), and steroleosin (~40 kDa). Oleosin and caleosin are the main structural proteins consisting of a central hydrophobic domain anchored inside the TAG, and a hydrophilic N-terminal and a C-terminal facing the cytoplasm [3,4,5,6,7]. Steroleosin is not considered to contribute to the structure stability of OBs, although it comprises a similar structure as caleosin, e.g., a shorter hydrophobic sequence and longer hydrophilic domains as compared to oleosin [8,9,10,11]. OBs can be obtained by aqueous extraction, through which the produced OBs are in the form of aqueous creams or emulsions. On one hand, the OB droplets are naturally emulsified without the addition of other surfactants or chemicals. On the other hand, natural nutrients such as fat-soluble vitamin E, and unsaturated fatty acids which are the naturally predominant components of the TAG molecules at the sn-2 position, can be completely preserved in such extracted natural OBs [4,11,12,13,14]. Moreover, the natural OB emulsion has a physical and chemical stability against external environmental perturbation, such as mechanical stresses, temperature, and oxidation [8,15], which is attributed to the protection by the phospholipid-protein membrane formed by the endogenous proteins and the outer layer stabilization by the exogenous proteins in the OB emulsion. The endogenous proteins on the surface of the OBs provide a certain charge and steric hindrance, resulting in a repulsive force that is relatively weak between the OB droplets and, thus, maintains the stability of the OB emulsion [16,17]. The exogenous proteins provide a second layer protection on the OB droplets by increasing the repulsive force between droplets and by the steric effect [18]. Therefore, the OBs and their aqueous emulsions have received extensive attention, and have the potential for developing beverages [19], edible films and coatings [20], and salad dressings, in the field of food industry [21].

In practical applications and development of food products using natural OBs, the stability of OB emulsions is affected by the processing conditions and the external environments, such as pH, freeze–thaw cycle, or salt ion strength, which might cause aggregation, coalescence, and demulsification [22,23]. The physical instability of OB emulsions can be improved by adding surfactants or biopolymers [16,18,21,23,24,25]. The surfactant can adsorb on the OB membrane or displace a surface protein, forming a denser membrane composed of surfactant and protein-phospholipids. In recent years, polysaccharides such as xanthan gum [18], gum arabic [24], carrageenan [23], and pectin [16,26] were used to improve the stability of OB emulsions. The charged polysaccharide can interact with the OB surface proteins, forming a second layer by improving the spatial repulsive force, reducing depletion flocculation, and improving the OB droplet stability [27]. As a linear anionic polysaccharide, sodium alginate (ALG) has unique functions, such as slowing down the absorption of fatty acids and bile salts, thus reducing the content of serum, cholesterol, blood triglycerides, and blood glucose in human body [28]. ALG can be dissolved and mixed at room temperature without heating, and can build up viscosity with a relatively low concentration. Therefore, the use of ALG to stabilize OB emulsion is highly significant and promising in the food industry. In a previous study, we stabilized the soybean OB emulsion by using ALG in different environments (pH, salt, and freeze–thaw cycle), and found that ALG can be adsorbed onto the surface of OB droplets, preventing OB aggregation at low pHs [29].

However, depending on the sources and extraction methods, the composition and amount of the surface proteins, as well as the exogenous proteins of diverse OBs might vary, therefore, affecting the properties and application of the OBs [1,8,29,30,31,32,33]. For example, it was observed that small maize germ OBs exhibited high stability against coalescence as they flocculated with other neighboring oil droplets in the aqueous environment, while larger-sized OBs, e.g., from other origins, might not be able to resist the mechanical stresses arising from depletion flocculation, thus, leading to coalescence and hence emulsion destabilization [18].

In this study, we extracted peanut, sesame, and rapeseed OB creams by the aqueous medium method. The moisture, fat, and protein contents of the extracted OB cream were analyzed. The surface protein composition, microstructure, average particle size $d_{4,\;3}$, *ζ*-potential, and stability of the OB emulsions were measured. Compared with the soybean OB in the previous study [29], the peanut, sesame, and rapeseed OB creams were composed of more than 55% fat, and their emulsions not only showed aggregation phenomenon near the isoelectric point (IEP), but also showed a creaming phenomenon at high pHs. In addition, the peanut, sesame, and rapeseed OB emulsions were severely affected by salt ions, as compared to the soybean OB emulsion. ALG was used to tackle the problem of the instability of the OB emulsions at different pH values by changing the concentration of ALG. The concentration of the ALG under different conditions was optimized; the mechanism is discussed herein. The salt and heat resistance of the OB emulsions stabilized by the optimized ALG concentration were also evaluated. Thus, this study broadened the range of applications for diverse OBs under different conditions.

## 2. Results and Discussion

### 2.1. OB Characteristics

The main chemical compositions of the three OB creams (peanut, sesame, and rapeseed) extracted using the aqueous media method are listed in Table 1. For comparison, the table also provides the compositions of the soybean OB cream from Su et al. [29]. Among the three OBs (peanut, sesame, and rapeseed), the average particle size of the sesame OB was the largest, and that of the rapeseed OB was the smallest. Their particle size distribution is shown in Figure 1a. The protein content of the rapeseed OB was the highest, and that of the sesame OB cream was the lowest, in contrast to the order of the fat content. Therefore, the rapeseed OB had the highest protein-to-fat ratio, while the sesame OB had the lowest among the three OBs. However, both the protein content and the protein-to-fat ratio for all three OBs were lower than those of the soybean OB, as listed in Table 1.

The *ζ*-potential of the peanut, sesame, and rapeseed OB droplets in emulsion at different pHs are shown in Figure 1b. It can be seen that the *ζ*-potential of the peanut, sesame, and rapeseed OBs all decreased from pH 3 to pH 8, and the IEPs were about pH 4.6, 4.0, and 4.9, respectively.

The surface protein species of the three extracted OBs were characterized by sodium dodecyl sulfate polyacrylamide gel electrophoresis (SDS–PAGE) (Figure 1c–e). The protein bands of the peanut OB are 18, 21, 25, 30, 40, 54, and 66 kDa, and larger Mws (Figure 1c). Among them, proteins of 18 kDa and 21 kDa are the endogenous proteins of the peanut oleosins, 25 kDa and 30 kDa are the caleosins, and proteins of 40 kDa are the steroleosins [24,34]. The other proteins with higher Mws (54 and 66 kDa) are the exogenous proteins [34,35]. The protein bands of the sesame OB are 14, 16, 21, 24, and 26 kDa (Figure 1d). All of which are the sesame oleosins (Mw of 14, 16, and 21 kDa), and the sesame caleosins (Mw of 24 and 26 kDa) [34,36], indicating that the exogenous proteins were almost completely removed. In Figure 1e, the proteins of the rapeseed OB are the endogenous oleosins with Mw of 17 kDa, caleosins with Mws of 25 and 30 kDa, and exogenous proteins of higher Mws of 54 and 88 kDa [34,37,38]. These results suggested that the OBs carried different endogenous and exogenous proteins. First, compared to the peanut and rapeseed OB, the sesame OB had substantially no exogenous proteins. Second, even for the same type of endogenous proteins, their Mw varied [32]. These differences were mainly due to the fact that the endogenous and exogenous proteins of different seed OBs were differently influenced by the same extraction conditions and cleaning agents [25,34]. The differences in the type and content of endogenous and exogenous proteins influence the stability of the OB emulsions since the structures such as the amino acid sequence and the number of the proteins are different. Therefore, we further investigated the stability of these OB emulsions at different conditions and used ALG to solve the stability problem in the following sections.

### 2.2. Influence of pH on the Creaming Stability of OB Emulsions

The effect of different pHs (from 3 to 8) on the stability of different OB emulsions was first examined. Photomicrographs of the OB emulsions at different pHs are displayed in Figure 2a–c. At pH 3, the peanut and rapeseed OB emulsions were well-dispersed, although the sesame OB emulsion showed a slight aggregation. At pH 4–6, all three OB emulsions showed a certain degree of aggregations. The sesame OBs were likely to have the largest aggregation among the three OBs. At pH 7 and 8, all OBs were well-dispersed in the emulsion, as they were strongly negatively charged. After storage for 7 days at room temperature, all three OB emulsions showed a creaming phenomenon at all pHs, as shown in Figure 2d–f. The creaming speed of the sesame OB emulsion was the fastest. The emulsions were extremely unstable near their IEP, which was probably due to the fact that the *ζ*-potential was low and the electric repulsion between the OB droplets was weak [22,29,39,40]. The OBs were prone to aggregate and then speeded up the rate of creaming. At a pH away from the IEP, the OB emulsions could maintain a certain stability due to the electrostatic repulsion [16] and steric hindrance of the surface proteins [4]. However, a completely stable OB emulsion over storage could not be obtained by using the extraction method employed in this study.

These results indicated that the stability of the OB emulsions was related to pH and was also affected by the amount of endogenous and exogenous proteins, as well as the nature of the oleosins, as has been discussed previously [40]. The contents of the endogenous and the exogenous proteins of the sesame OBs were less than that of the other two seed OBs, according to Figure 1c–e. The sesame OBs had almost no exogenous proteins and a very small amount of endogenous proteins. Thus, the amount of charges provided by the sesame OB surface proteins was lower than those of peanut and rapeseed OBs, and the IEP of the sesame OB moved to the direction of a lower pH [40]. In addition, the primary structure of the sesame oleosins showed that the sesame oleosins had shorter hydrophilic terminals on the surface of the sesame OBs [40,41,42]. Therefore, the stability of the sesame OB emulsion was worse than the other two OBs, due to the insufficient electrostatic repulsion and low steric protection. In the creaming process of the OB emulsions, the OB droplets diffused and rose up to the air–water surface, causing the increase of the interfacial pressure and, thus, some OBs at air–water surface might rupture, although others remain integrated and agglomerated [43]. Therefore, aggregation and creaming of OB emulsions affect the processing and quality of the food products, and, should be intervened.

### 2.3. Influence of ALG on OB Emulsions

The OB emulsions had the most serious aggregation and creaming near their IEP, and were also very unstable during storage where aggregation occurred when the ion concentration or temperature changed [23,29,44]. It has been shown that anionic polysaccharide ALG or alginate type hydrocolloids (pectin) could be adsorbed on soybean OB surface at a pH lower than IEP, which increased the stability of the OBs by reducing the van der Waals attraction and increasing the electrostatic repulsion [17,29,39]. In this work, the ALG was applied to stabilize the unstable peanut, sesame, and rapeseed OB emulsions at a pH around their IEP. First, the ALG concentrations were optimized. A pH slightly lower than the IEP of each OB emulsion was chosen to find the optimum concentration of ALG—pH 3.9 was chosen for the sesame and peanut OB emulsions; pH 4.5 was chosen for the rapeseed OB emulsion. The effect of ALG concentrations on the *ζ*-potential and particle size of the three OB emulsions is shown in Figure 3. In Figure 3a–b, at the experimental pH of each OB emulsion, as mentioned above—the *ζ*-potential of the OB emulsions and the particle size decreased with an increasing concentration of ALG. When the ALG reached a certain concentration, the *ζ*-potential and particle size basically reached a plateau value, which meant that the minimum concentration of ALG which could stabilize the OB emulsion was reached. The stabilized OB emulsion with this concentration of ALG had a relevant large *ζ*-potential, and the smallest particle size, since the electrostatic repulsion between oil particles was strong enough to keep the stability of droplets against aggregation. Therefore, the optimal concentration of ALG was 0.35 wt.% for the peanut OB emulsion, 0.45 wt.% for the sesame OB emulsion, and 0.3 wt.% for the rapeseed OB emulsion (emulsions all with 1 wt.% OB cream). The concentration of saturation adsorption of ALG was inversely related to the IEP of different OBs, which was determined by the structure of the oleosins [40].

The creaming stability of the OB emulsions with the optimum concentration of ALG was then investigated, and the results are displayed in Figure 4a–c. It can be seen that at a pH close to the IEP (pH 3–6), the creaming stability of the three OB emulsions was relatively good, and much improved compared to that in the absence of ALG (comparing Figure 4a–c with Figure 2d–f), suggesting that ALG can effectively balance the charge between the surface proteins of OBs, and further appropriately increase the repulsive force between OB droplets. Therefore, the OB can exist stably against aggregation at these pHs in the emulsion.

However, the creaming phenomenon for all of the three OB emulsions still markedly occurred at pH 7 and 8, as shown in Figure 4a–c. Conversely, in the previous study, soybean OB emulsion stabilized by 0.35 wt.% ALG at pH 7 and 8 could evenly disperse and the creaming phenomenon did not occur [29]. As mentioned above, the creaming and destabilization of the OB emulsion was related to the diffusion and exposure of the OBs to the water–air–interface [40,43]. There might be several reasons for the creaming stability at pH 7 and 8 for the soybean OBs with low concentration ALG. First, the soybean OB emulsions contained more proteins on the OB surfaces, with the structure proteins of oleosin and caleosin being rich in content, which led to a better stability than the other three OB emulsions studied here [4,21]. Furthermore, the soybean OBs had the smallest particles size among the four OBs (Table 1), which slowed the creaming speed of the oil droplets in the emulsion during storage. Therefore, a low concentration of ALG could stabilize the soybean OB emulsion. In the present study, the droplet sizes in the sesame or peanut OB emulsions (2.31 ± 0.12 μm or 3.65 ± 0.01 μm, respectively as shown in Figure 1a) were much larger than that of the soybean OBs (about 0.54 ± 0.01μm), therefore, the creaming speed was probably faster. For the rapeseed OBs, although their droplet sizes were also small, they had less oleosin and caleosin on the surface than the soybean OBs, and the surface charge was also low at pH 7, resulting in a smaller repulsive force or steric hindrance between the OBs and, thus, it was easier for the OBs to aggregate and float up. Therefore, a low concentration of ALG could not stabilize all three OB emulsions at pH 7 and 8 in the present study.

### 2.4. Influence of ALG on the Creaming Stability of OB Emulsions at pH 7

In order to stabilize the OB emulsion at pH 7 and 8, and slow down the creaming phenomenon, the concentration of the ALG added to the OB emulsions was increased. The stability of the OB emulsions with different concentrations of ALG under neutral conditions after 7 days of storage was analyzed, and the results are displayed in Figure 5a–d. When the ALG concentration was 0–0.9 wt.%, the OB emulsion showed obvious creaming, that is, the OBs floated up, resulting in a cream layer on the top of the emulsion and a serum layer at the bottom, as shown in the “0%” sample on the left in Figure 5a–c (data for other ALG concentration sample are shown in Appendix A). The creaming indices for the cream layer and the serum layer of the OB emulsions, as defined in Section 3.5, were high in these samples (Figure 5d). As the concentration of ALG increased, the creaming of OB emulsions was gradually slowed down, and both the serum and creaming indices were reduced. When the ALG concentration reached 1.2 wt.%, no creaming phenomenon was observed in the sesame OB emulsion. When the ALG concentration reached 1.5 wt.%, the creaming phenomenon of the peanut and rapeseed OB emulsions did not occur.

To understand the mechanism of ALG stabilizing OB emulsions, Figure 6a–c show the viscosity of the pure OB emulsion at pH 7 and the OB emulsions with different concentrations of ALG as a function of the shear rate. The results showed that all three 1 wt.% of pure OB emulsions exhibited shear thinning characteristics within the shear rate range examined. However, when the shear rate was between 10–100 s^−1^ the viscosity of these emulsions had stable values, which were about 2–4 mPa·s. When the ALG concentration increased to the optimal concentration for stabilizing the emulsions at around pH 4, which was 0.35 wt.%, 0.45 wt.%, and 0.3 wt.% ALG for the peanut, sesame, and rapeseed emulsions, respectively, as discussed in Section 2.3, the viscosities of the OB–ALG emulsion systems were increased by about 4–7 folds higher than each pure OB emulsion between 10 s^−1^ and 100 s^−1^. Whereas, when the ALG concentration was increased to 1.2 wt.% for the sesame OB emulsion and 1.5 wt.% for the peanut and rapeseed OB emulsions, the viscosities of the OB emulsions were significantly increased, e.g., the viscosity of each was about 200–700 mPa·s between 10 s^−1^ and 100 s^−1^, which was more than 100 times higher than the pure OB emulsions. According to Stokes formula [45], the increase of viscosity could increase the drag force exerted on the droplets and reduce their creaming rates in the emulsion. Meanwhile, the *ζ*-potential of low concentration ALG and high concentration ALG stabilized OB emulsion were measured (Appendix A). The OB emulsions with different ALG concentrations (0.30 wt.%, 0.35 wt.%, 0.45 wt.%1.20 wt.%, and 1.50 wt.%) showed similar *ζ*-potential at pH 7. Therefore, we believed it was the viscosity effect that effectively controlled the creaming phenomenon of the OB emulsions at pH 7 and 8, since the increase in the concentration of ALG increased the emulsion viscosity. The addition of ALG at a lower concentration did not solve the creaming phenomenon of the OB emulsions at pH 7, since the viscosity of each system was still not high enough to slow down the creaming of OB droplets, whereas the high viscosity of the high ALG concentration successfully slowed down the movement of the OBs.

### 2.5. Influence of Salt on the Stability of OB Emulsions

During the food processing, the concentration of salt ions could influence the conformation of the surface proteins and the exogenous proteins of OBs, thus, affecting the stability of OB emulsions. Therefore, the stability of the OB emulsion under different NaCl concentrations was investigated at different pHs (pH 4 and 7, were selected as the typical acidic and neutral conditions). The surface charge and particle size of the different OB emulsions with varying NaCl concentrations are shown in Figure 7a–c and Figure 8a–c, respectively. It could be seen that the *ζ*-potential of the pure peanut and sesame OB emulsion were between −2 mV and 2 mV at pH 4, while the *ζ*-potential of the rapeseed OB emulsion at pH 4 was about 9 mV. Their *ζ*-potentials did not change much with the increase of salt ion concentration (Figure 7a–c). Iwanaga et al. mentioned similar results in their studies on the influence of NaCl on OBs [22]. They attributed the constancy of *ζ*-potential with the increase of salt concentration to the presence of endogenous salt or the charge regulation effect of the OBs. When exogenous salt (NaCl) was added to the OB emulsion system, there was an electrostatic screening effect. However, the addition of exogenous monovalent cations (Na^+^) might partially displace any divalent cations (Mg^2+^ or Ca^2+^) associated with the anionic OB surface, thereby, counterbalancing the expected decrease in negative charge and weakening the electrostatic screening effect, such that the expected decrease of *ζ*-potential would be counterbalanced. Since the charges on the surface of the particles at pH 4 are still low, the OBs tend to aggregate to larger particles to a certain extent, as indicated in Figure 8a–c. The particle size change of the rapeseed OB emulsion was particularly obvious, as its particle size increased from 7.8 μm (0 mmol/L NaCl) to 12.3 μm (250 mmol/L NaCl), as displayed in Figure 8c, which was about a 57% increase. However, the particle size of the peanut and sesame OB emulsions did not change obviously, since pH 4 was close to the IEP of the peanut or the sesame OB emulsion, at which the OBs aggregated more seriously than the rapeseed OBs, as shown by the large particle size in Figure 8a–b and the microstructure in Appendix A. Therefore, the electrostatic screening effect and charge regulation did not obviously affect the particle size of the peanut and the sesame OB emulsions.

When ALG was used to stabilize the OB emulsions at pH 4, as shown in Figure 7a–c, the *ζ*-potential of the three OB emulsions became largely negative, with values of around −30 to −40 mV, not changing significantly with the NaCl concentration. In contrast to the pure OB emulsions, the particle size decreased significantly, especially for the peanut and sesame OB emulsions with ALG (Figure 8a–c). Additionally, the particles size for each OB–ALG emulsion did not change with the NaCl concentration, indicating that the emulsions were dispersed uniformly and were stable against NaCl.

The stability of the OB emulsions, with and without ALG, at different salt ion concentrations at pH 7 was also measured. The results are displayed in Figure 7a–c and Figure 8a–c. The *ζ*-potential of the pure peanut and sesame OB emulsions were about −10 mV to −20 mV, without any significant variation, while *ζ*-potential of the rapeseed OB emulsion showed an upward trend from −13.93 mV (0 mmol/L NaCl) to −10.27 mV (250 mmol/L NaCl). In Figure 8a–c, with an increase of the NaCl concentration, the particle size of the peanut OB emulsion was around 3–5 μm, and that of the sesame OB emulsion was around 4–5 μm, implying no obvious change. The peanut and sesame OB emulsions were rich in structural proteins. The particle size of the rapeseed OB emulsion increased a little as the NaCl concentration increased, probably also due to the electrostatic screening effect caused by the addition of Na^+^. At this pH (7), when a high concentration of ALG was added to each of the OB emulsions, as we optimized in Section 2.4, all ALG-stabilized OB emulsions had an enlarged charge density (Figure 7a–c) and a decreased particle size (Figure 8a–c), which did not change markedly with an increase in NaCl concentration. Therefore, a high concentration of ALG could also improve the stability of the OB emulsion against the salt ions.

### 2.6. Influence of Freeze–Thaw Cycling and Thermal Treatment on the Stability of OB Emulsions

The effect of freeze–thaw cycling on the stability of the OB emulsions was also investigated. Similarly, the freeze–thaw stability under acidic and neutral conditions was examined, in terms of particle size, microstructure, and creaming stability. As can be seen from Figure 9a–c, the particle size of the pure OB emulsions increased with the increase of the number of freeze–thaw cycles at pH 4, especially for the peanut and sesame OB emulsions. This was mainly because the three OB emulsions were in an unstable state at pH 4, and the oil droplets coalesced after freezing and thawing, forming large particle droplets. The creaming index for the serum layer of the freeze–thawed emulsion at pH 4 (Figure 10a–c), was also significantly high, indicating the instability of the emulsion against freeze–thaw cycling.

At pH 7, all pure OB emulsions showed an even increase of particle size after freeze–thaw treatment, following obvious coalescence of oil droplets and demulsification, although the result after the third freeze–thaw cycle might be unreliable, since the coalescent oil droplets were easily attached to the sample vial. The real particles sizes were even greater than the data shown in Figure 9a–c for pure OB emulsions after three times of freeze–thaw cycling at pH 7. The creaming index in Figure 10a–c also show a serious creaming phenomenon [44,46,47,48].

After addition of ALG at the optimal concentration to either pH, the particle size of the peanut, sesame, and rapeseed OB emulsions all became much smaller, as compared to the pure OB emulsion, frozen and thawed by the same number of freeze–thaw cycling. Even after three cycles of freezing and thawing, the peanut, sesame, and rapeseed OB emulsions stabilized by ALG showed a slight increase of particle size, respectively, and the demulsification phenomenon did not occur any more. The creaming phenomenon after freezing and thawing was also controlled, and no obvious creaming occurred in these ALG-stabilized emulsions with a very low creaming index obtained in Figure 10a–c. Therefore, the addition of ALG protected the OB emulsion against the freeze–thaw cycling process, and improved the stability of the OB emulsions at both acidic and neutral conditions.

Effect of thermal treatment at high temperatures (60, 90 and 120 ℃) on the stability of the OB emulsions was also investigated. The results were displayed in the Appendix A. After thermal treatment at each temperature for 30 min, the ζ-potential (Appendix A) of each OB emulsion did not change much compared with that after heating at 25 ℃. The microstructure (Appendix A), and particle size and its corresponding change (Appendix A) after high temperature treatment of the three pure and ALG-stabilized OB emulsions at pH 4 or pH 7 are also similar to those at 25 ℃. The results indicate that ALG stabilized-OB emulsions also have great stability against high temperature thermal treatment.

## 3. Materials and Methods

### 3.1. Materials

Peanuts and sesame were purchased from Shenyang Xinchang Grain Trade Co. Ltd. (Shenyang, China). Rapeseed was provided by the Institute of Oil Crops, Chinese Academy of Science. The ALG was provided by FMC BioPolymer (Drammen, Norway), which had a molecular weight (Mw) of 270 kDa and a polydispersity index of 1.50. Other chemicals were of analytical grade and were purchased from the Sinopharm Chemical Reagent Co. Ltd. (Shanghai, China). Ultra-pure water (LBS-RUP60, Chengdu Haokang Technology Co., Ltd., Chengdu, China) was used for the preparation of all solutions.

### 3.2. OB Extraction

OBs were extracted from plant seeds (peanut, sesame, and rapeseed) according to the method described by Su et al. [29]. Generally, the soaked seeds were blended by a commercial food processor (KS-920, Guangzhou City Electric Appliance Co., Ltd., Guangzhou, China) to obtain a homogenate of each. Then, the homogenate was filtered and the filtrate was centrifuged at a high acceleration of 10,000× *g*, for 30 min at 4 °C (CR21N, Hitachi Koki Co., Ltd., Tokyo, Japan), and was washed once with urea and with Tris-HCl buffer solution for three times.

### 3.3. Characterization of OB Chemical Compositions

The chemical compositions (the contents of fat, proteins, and moisture) of the extracted OB cream were determined according to the Association of Official Analytical Chemists (AOAC) methods [29,49]. The fat content of the OBs was determined by the Soxhlet extractor system using petroleum ether as the extraction solvent. Nitrogen content of the OBs were determined by the Kjeldahl method. The 6.25× Kjeldahl N conversion factor was used to convert the percentage of nitrogen to the protein content. The moisture content of the OBs was determined by oven drying.

The composition of the OB surface proteins was analyzed by SDS–PAGE, according to the method described by Su et al. [29].

### 3.4. Preparation of OB Emulsions with Different Concentrations of ALG

The peanut, sesame, and rapeseed OB emulsions (5 wt.%) were prepared by dispersing 5 g of each extracted OB cream in 95 g PBS (50 mmol/L, pH 7), respectively, under stirring for 2 h, followed by sonicating for 3 min (Frequency, 20 kHz; Amplitude, 40%; Duty cycle, 1 s), using a high-intensity ultrasonic probe device (VCX800, 53 Church Hill Rd. Newtown, CT, USA) [29].

ALG solution (3 wt.%, pH 7) was prepared by dispersing the weighed powders of ALG into PBS (50 mmol/L, pH 7) under magnetic stirring for 24 h. Then, 20 mL of each OB emulsions were mixed with a different amount of ALG, respectively, to achieve the desired concentration. And then the mixtures were stirred for 2 h at 22 ± 2 °C. Finally, the pH value of the OB emulsions with ALG was adjusted, using 1 mol/L HCl, under magnetic stirring for another 30 min. The samples were kept at 22 ± 2 °C for 24 h before analysis.

### 3.5. Particle Size, ζ-Potential, Microstructure, and Emulsion Stability Measurements

The particle size of diluted OB emulsions and OB emulsions with ALG was measured using a laser light scattering instrument (Malvern Mastersizer 2000, UK) [29]. The OBs were diluted with water to an approximate oil concentration of 0.005 wt.%. The refractive index used for OB was 1.47, and that of water was 1.33. The density of OB used was 0.92 g/cm^3^. The volume-weighted mean diameter ($d_{4,3}=\sum n_{i}d_{i}^{4}/\sum n_{i}d_{i}^{3}$) was measured and used in the following analysis, where $n_{i}$ was the number of droplets of diameter $d_{i}$.

The electric potential (*ζ*-potential) of the OB emulsion was measured by diluting the sample to an approximate oil concentration of 0.005 wt.% (diluted with PBS) and was calculated by the electrophoretic mobility of the droplets measured, using a capillary electrophoresis cell (Zetasizer Nano ZS series, Malvern Instruments, Worcestershire, UK), expressed as the average values from triplicate measurements [29].

An Olympus microscope (Olympus IX73, Olympus Corporation, Tokyo, Japan) mounted with a scientific CMOS camera (Prime, Teledyne Photometrics, Surrey, Canada) was used to observe the microscopic morphology of the OB emulsions, which were diluted with PBS at different pHs, salt ionic strength, and after freeze–thaw cycling or high temperature thermal treatment. A small amount of each OB emulsion sample was placed on a glass slide and a cover glass was applied gently to form a thin sample layer and the slide was placed under the microscope with “4×” and “20×” objective for visualization.

The creaming stability of the OB emulsion was evaluated at 22 ± 2 °C, according to the method of Wu et al. [23]. A total of 10 mL of each emulsion was placed in a sealed glass vial for storage. After 7-days of storage, the stability of the emulsion was indicated by the creaming indices, defined as—the creaming index of the serum layer (%) = (height of serum layer/total height of the emulsion) × 100%, and the creaming index of the cream layer (%) = (height of cream layer/total height of the emulsion) × 100%, respectively. The creaming stability of the emulsion was controlled or there was no creaming instability for the emulsion if its creaming index of the serum layer and that of the cream layer were both 0%.

The influence of pH, salt ionic strength, freeze–thaw cycling, and thermal treatment on pure OB emulsion and OB emulsions stabilized by ALG were examined. The emulsions were adjusted to a different pH (pH 3, 4, 5, 6, 7 and 8), using 1 mol/L HCl or NaOH. The OB suspensions with different ionic strengths (0–250 mmol/L) were prepared by adding different ratios of the sodium phosphate buffer (50 mmol/L, pH 7) and 1 mol/L NaCl. For the freeze–thaw cycling and thermal treatment, a quantity of 10 mL of each OB emulsions, uncoated and coated with ALG at pH 4 or pH 7, was transferred in a sealed glass bottle. This freeze–thaw cycling was repeated up to three times. These emulsions were stored at a low temperature of −20 °C for 22 h, and then the suspensions were heated at 40 °C for 2 h. For thermal treatment, the OB emulsions were heated at a fixed temperature (25, 60, 90 and 120 ℃) for 30 min, respectively, and then the properties of the OB emulsions including *ζ*-potential, particle size, and microstructure were examined after cooling them to ambient temperature (25 °C).

### 3.6. Viscosity Measurements of OB Emulsions

The viscosity of the different OB emulsions with ALG was measured by using a HAAKE RS6000 Rotational Rheometer with P35TiL parallel-plate. The gap was set to 0.50 mm. About 0.50 mL OB emulsions with a low and high concentration of ALG, which were prepared as described in Section 2.4, were placed between the plates. The excess sample was removed and silicone oil was covered at the periphery of the exposed portion of the sample to prevent evaporation of water. The measurements were performed at a constant temperature of 25.0 °C ± 0.1 °C. The shear rate was increased from 10 s^−1^ to 500 s^−1^, and the viscosity was recorded as a function of the shear rate.

### 3.7. Data Analysis

Each emulsion sample was prepared at least three times, and each measurement of emulsion physicochemical properties was conducted at least in triplicates. Therefore, at least 9 analyses were done for each parameter of a sample, and the statistics were calculated to get the average value with the standard deviation.

## 4. Conclusions

This study showed that the peanut, sesame, and rapeseed OBs extracted by the aqueous extraction method have different types and contents of endogenous and exogenous proteins. At acidic conditions near the IEP of their emulsions, 0.35 wt.%, 0.45 wt.%, and 0.30 wt.% ALG could stabilize the 1 wt.% peanut, sesame, and rapeseed OB emulsions, respectively. At this condition, ALG molecules adsorb onto the surface of the OBs through electrostatic interaction, balance a certain positive charge, and enhance the repulsion between the OB droplets, stabilizing the emulsions. For different OBs, the optimum ALG concentration was different, which might be related to the surface protein types and contents of the extracted OBs. These low-concentration ALG-coated OB emulsions showed better storage, salt tolerance, and freeze–thaw resistance than the pure OB emulsions. However, a serious creaming phenomenon was observed in the emulsions at pH 7–8. By increasing the concentration of ALG, we successfully solved this problem, and the creaming stability of the OB emulsions was much improved through the viscosity effect. The salt tolerance, temperature stability, and freeze–thaw resistance of the OB emulsions were also significantly improved by adding the high concentration ALG to this neutral condition.

## Figures and Tables

**Figure 1 molecules-24-03856-f001:**
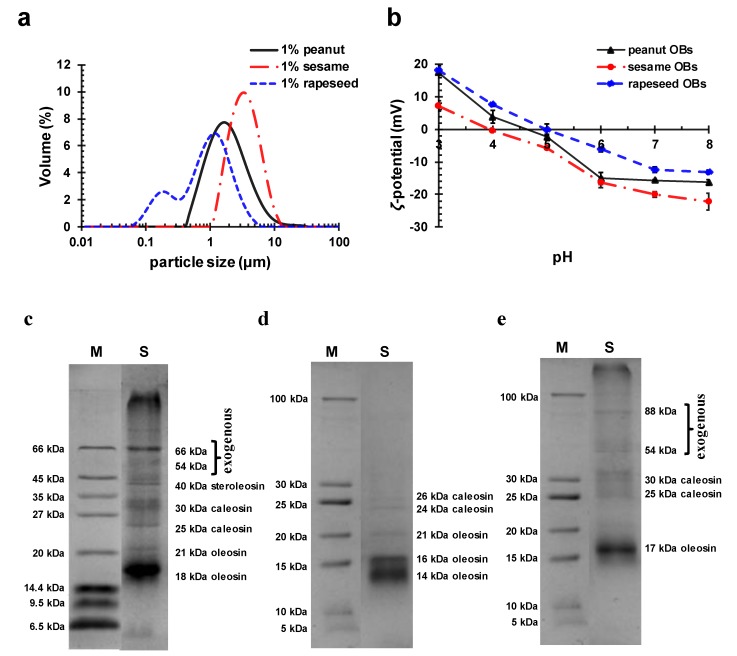
(**a**) The particle size distribution at pH 7, (**b**) the *ζ*-potential at various pHs of the three oil body (OB) emulsions (1 wt.%), and SDS–PAGE of the (**c**) peanut, (**d**) sesame, and (**e**) rapeseed OB surface proteins. Column M—molecular mass markers; and column S—proteins on the surface of the OBs.

**Figure 2 molecules-24-03856-f002:**
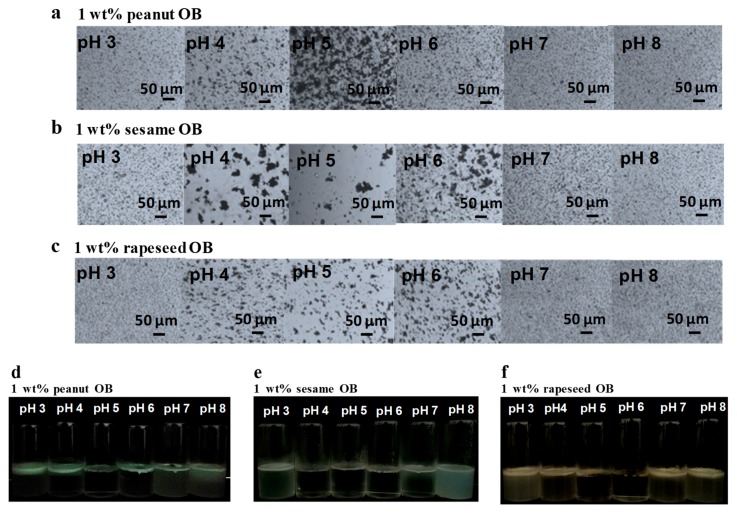
pH effect on the microstructure and creaming stability of OB emulsions during storage—(**a**,**d**) peanut, (**b**,**e**) sesame, and (**c**,**f**) rapeseed OBs. The OB creams were dispersed in 50 mmol/L sodium phosphate buffer solution (PBS) to a concentration of 1 wt.%. The creaming observation was made after storage at 22 ± 2 °C for 7 days.

**Figure 3 molecules-24-03856-f003:**
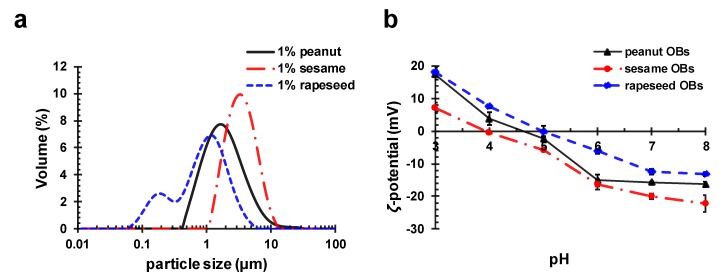
(**a**) The *ζ*-potential and (**b**) particle size $\left(d_{4,3}\right)$ of the three OBs coated by different concentrations of sodium alginate (ALG) at a pH of about 4 (pH 3.9 for the sesame and peanut emulsions; pH 4.5 for the rapeseed emulsion).

**Figure 4 molecules-24-03856-f004:**
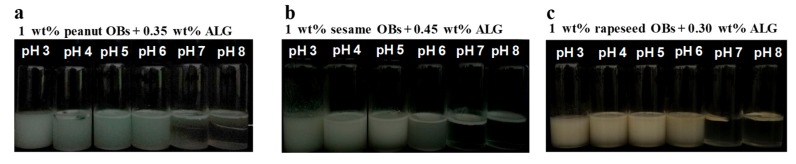
pH effect on the creaming stability of ALG-coated OB emulsions (**a**) peanut (1 wt.% OB and 0.35 wt.% ALG), (**b**) sesame (1 wt.% OB and 0.45 wt.% ALG), and (**c**) rapeseed (1 wt.% OB and 0.30 wt.% ALG). The OBs were dispersed in 50 mmol/L PBS. The creaming observation was made after storage at 22 ± 2 °C for 7 days.

**Figure 5 molecules-24-03856-f005:**
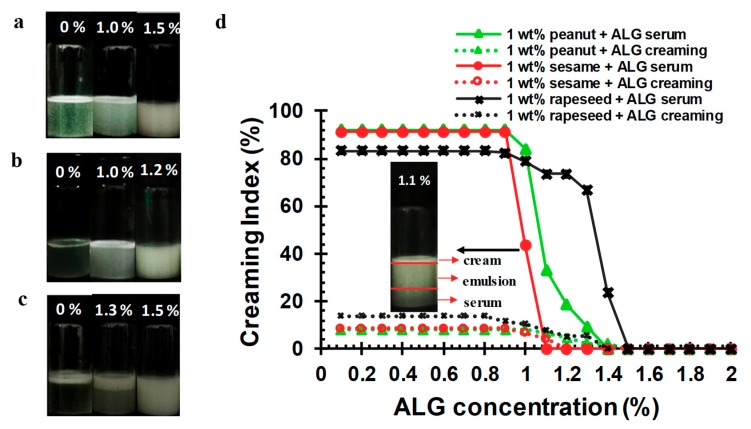
The creaming stability of (**a**) peanut, (**b**) sesame, (**c**) rapeseed emulsions and the creaming index (**d**) of the three OB emulsions stabilized by different concentrations of ALG at pH 7. The percentage concentrations in Figure a, b, c represent the concentration of ALG. The creaming investigation was made after storage for 7 days at 22 ± 2 °C.

**Figure 6 molecules-24-03856-f006:**
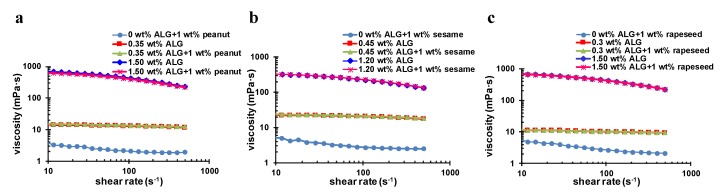
The viscosity of (**a**) peanut, (**b**) sesame, and (**c**) rapeseed OB emulsions in the presence of ALG with different concentration as indicated (at pH 7).

**Figure 7 molecules-24-03856-f007:**
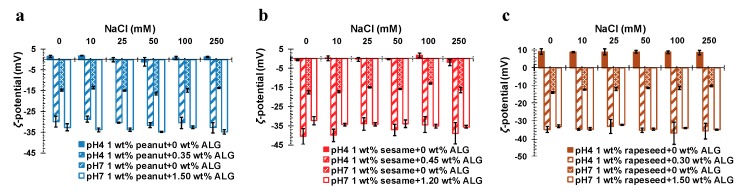
Salt effect on *ζ*-potential of pure and ALG stabilized OB emulsions at different pHs as indicated: (**a**) peanut, (**b**) sesame, and (**c**) rapeseed. The OBs were dispersed in 50 mmol/L PBS.

**Figure 8 molecules-24-03856-f008:**
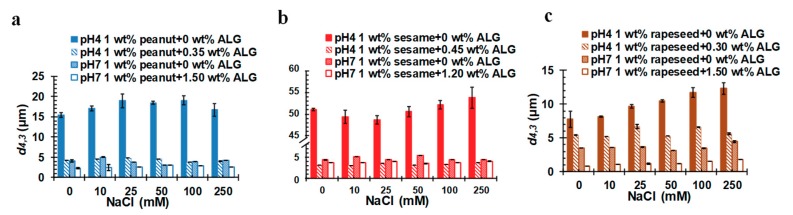
Salt effect on mean particle diameter ($d_{4,3}$) of pure and ALG-stabilized OB emulsions at different pHs as indicated—(**a**) peanut, (**b**) sesame, and (**c**) rapeseed. The OBs were dispersed in 50 mmol/L PBS.

**Figure 9 molecules-24-03856-f009:**
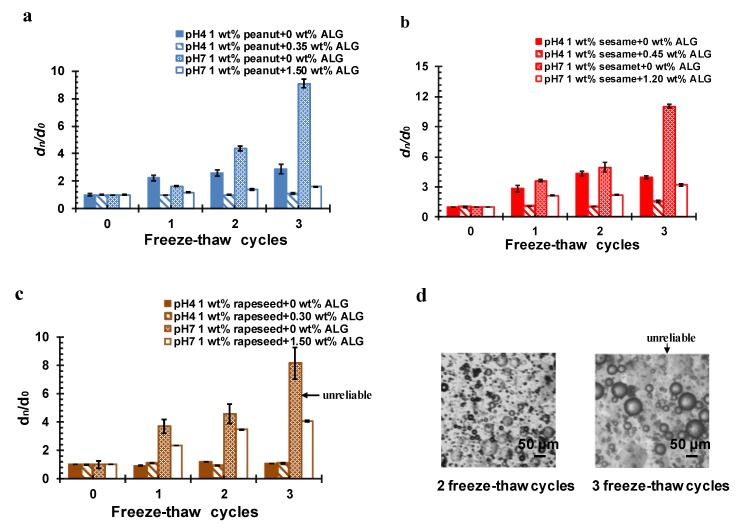
Effect of the freezing–thawing cycle on relative particle diameter ($d_{n}$/$d_{0}$) of pure and ALG-stabilized OB emulsions at different pHs, as indicated—(**a**) peanut, (**b**) sesame, and (**c**) rapeseed, (**d**) effect of the freezing–thawing cycles on the microstructure of rapeseed OB emulsions. The OBs were dispersed in 50 mmol/L PBS. $d_{n}$ was the droplet size after ‘n’ freeze–thaw cycles and $d_{0}$ was the droplet size before the freeze–thaw cycles.

**Figure 10 molecules-24-03856-f010:**
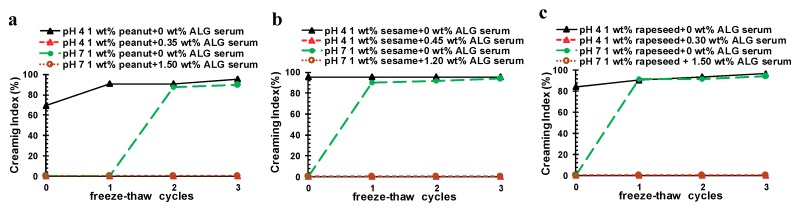
Effect of freezing–thawing cycle on the creaming stability of pure and ALG-stabilized OB emulsions at different pHs, as indicated—(**a**) peanut, (**b**) sesame, and (**c**) rapeseed. The OBs were dispersed in 50 mmol/L PBS. The creaming index here is for the serum layer of the OB emulsions after different freezing–thawing cycles.

**Table 1 molecules-24-03856-t001:** Chemical compositions of the extracted oil bodies (OBs) creams.

	$d_{4,3}(\mu m)$	Fat (wt.%)	Protein (wt.%)	Moisture (wt.%)	Protein/Fat
Soybean [29]	0.54 ± 0.01	29.40 ± 1.00	3.00 ± 0.20	53.60 ± 2.30	0.10 ± 0.01
Peanut	2.31 ± 0.12	72.91 ± 1.03	1.16 ± 0.01	22.72 ± 0.14	0.02 ± 0.01
Sesame	3.65 ± 0.01	80.56 ± 0.80	0.95 ± 0.01	17.66 ± 0.19	0.01 ± 0.01
Rapeseed	1.05 ± 0.01	55.34 ± 0.75	2.95 ± 0.04	37.38 ± 0.26	0.05 ± 0.01

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
