# Peer review of "Improving the Stability of Oil Body Emulsions from Diverse Plant Seeds Using Sodium Alginate"

_molecules, 2019, doi:10.3390/molecules24213856_

Round 1

Reviewer 1 Report

The present manuscript discusses the stability of oil bodies from different plant seeds such as sesame, peanuts, and rapeseeds. Their stability and creaming properties are compared and discussed. Moreover, the stability including freeze-thaw properties has been also investigated by the presence of sodium alginate. The results warrant publication.

However, the discussion and the interpretation lacks somehow of a conclusive picture. While the authors mainly discuss “what has been seen”, the experiments offer plenty of connections to the physical and chemical properties of oil bodies. The discussion of the results can be improved and made more solid.

1) The nature of the oleosins play a significant and universal role in the stability of the oilbodies as has been discussed recently by Maurer et al. (2013) The journal of physical chemistry B, 117(44), 13872-13883. Indeed the unique properties of the oleosins and their pH depending charge show clearly why the alginate adsorption changed respectively with pH-changes. It also shows clearly the different creaming behavior of the oil bodies. The slightly different primary structure of sesame oleosins close to the C- and N-terminal explain their different behavior (as well as their IEP at lower pH)

2) The creaming and destabilization seems to be related to the water method and the exposure of the oil bodies to water-air-interfaces as discussed Waschatko et al (2012). The Journal of Physical Chemistry B, 116(35), 10832-10841. If so, this point should be made clear.

3) Figures 3 and 6. show nicely how alginate saturates the “coating” (encapsulation) which is in accord with these ideas.

4) The behavior of the size of (soybean) oil bodies before and after encapsulation with alginate type hydrocolloids (pectin) have been measured in a direct way by small angle neutron scattering Zielbauer, B. I., et al. (2018), Journal of colloid and interface science, 529, 197-204.

5) The role of the salt (NaCl) should be made clearer. It looks like the electrostatic Coulomb screening takes the main role for the explanation (around Figure 8). The authors also suggest replacement of endogenous bivalent calcium ions by Na. This process should be explained in more detail.

6) Figures 6, 8, and 10 appear far to small, the captions and the explanations are very hard to read.

Reviewer 2 Report

In the present study, the authors extracted peanut, sesame and rapeseed seed oil bodies for stability study. The study basically followed a previous study of soybean oil bodies. Similar results were obtained for the improvement of oil-body stability by sodium alginate. Although the data were lack of novelty, they did provide useful information to broaden the range of applications for diverse seed oil bodies.

A major concern should be answered for the consideration of publication.

In Figure 9, freezing-thawing cycling led to the drastic increase of average sizes of seed oil bodies. The authors explained that the increase of oil-body sizes was due to coalescence (loss of oil-body integrity after particle fusion). The explanation seems to be reasonable. In Figure 3b, the average sizes of sesame oil bodies dropped from 37 μm to 14 μm after adding 0.05% sodium alginate, and further dropped to 7 μm in the presence of 0.2% sodium alginate. As sesame oil bodies are solid emulsions fulfilled with oil molecules (mainly triacylglycerols), it is unlikely that sesame oil bodies are able to shrink drastically (several times) after adding 0.05% or 0.2% sodium alginate without losing its integrity. The authors are suggested to explain this drastic observation scientifically.

Reviewer 3 Report

This manuscript deals with the stability of oil body emulsions from the peanut, sesame and rapeseed seeds by sodium alginate. 

The manuscript can not be accepted for publication in Molecules journal because it is mainly based on experimental data that extend published findings without adding substantial knowledge. Results are predictable.

Importantly, authors did not optimize the extraction process for the different seeds (e.g. salt addition). By using only water as the solvent, they recovered  a low amount of exogenous proteins (sesame seed).

They did not study thermal treatment with regard to OB stability although this treatment is important for the exploitation of OBs as raw material for different applications. 

Round 2

Reviewer 2 Report

The answer to my major concern is reasonable. Thus, the revised manuscript is acceptable.

Reviewer 3 Report

Authors made a significant effort to improve the quality of the work. However, the effect of the thermal treatment (60-120 oC) on the physical and chemical stability of the oil bodies should be presented and discussed to strengthen novelty of the content of the submitted manuscript, future applications in different food systems and thus to support publication in Molecules  journal.